



# Comment on "Weibel instability in a plasma with nonzero external magnetic field" by O. A. Pokhotelov and M. A. Balikhin, Ann. Geophysicae, 30, 1051–1054, (2012)

Rebiha Naceur[1], Abdelaziz Sid[1], Roumaissa Malim[1], Narges Firouzi-Farashbandi[2]

[1]Laboratory of Physics of Radiations and their Interactions with Matter, University of Batna 1, 05000, Batna, Algeria
[2]Department of Physics Education, Farhangian University, Taheran, Iran.

*Correspondence to*: Abdelaziz Sid (abdelaziz.sid@univ-batna.dz)

**Abstract.** In the reference (Ann. Geophysicae, 30, 1051–1054, (2012)), the dispersion relation of Weibel modes is explicitly calculated in magnetized plasma. The parameters of Weibel instability which are the Weibel mode frequency, $Re(\omega)$, and the growth rate of the Weibel instability, $Im(\omega)$ are deduced from the obtained dispersion relation under justified approximations.

Several errors, which affect the results, are identified in this reference (Ann. Geophysicae, 30, 1051–1054, (2012)). We are interested in the present work to correct these errors.

It has been shown that $Im(\omega)$ remains the same that in the case of zero external magnetic field. However, $Re(\omega)$, is proportional to the electron cyclotron frequency, $\sim \mp \omega_c$ , and to the temperature anisotropy, $\sim \frac{\Delta T}{T_\parallel} = \frac{T_\perp - T_\parallel}{T_\parallel} = \left(\frac{T_\perp}{T_\parallel} - 1\right)$, where $T_\perp$ and $T_\parallel$ are respectively the perpendicular and the parallel temperature.

Keywords: Weibel instability, Magnetized plasma, bi-Maxwellian distribution

## 1 Introduction

Erich S. Weibel (Weibel, 1959) was the first to demonstrate that temperature anisotropy generates electromagnetic instable modes in plasma. Since the appearance of the original Weibel paper ((Weibel, 1959), several studies have been carried out on Weibel instability.

In their paper (Ann. Geophysicae, 30, 1051–1054, (2012)), Pokhotelov and Balikhin investigate Weibel instability (Weibel, 1959) in magnetized plasma, assuming a standard bi-Maxwellian distribution function. The dispersion relation is derived in the noncollisionnel regime from the perturbed Vlasov equation combined with the Maxwell's equation. By using the quasi-static Weibel modes approximation, which permits to neglect the displacement current in Ampère's law and the higher-order terms, relative to the scale parameter $\epsilon = \left|\frac{\omega \pm \omega_c}{v_{T\parallel}k}\right|$, the authors are explicitly calculated the growth rate of Weibel mode and the Weibel mode frequency.





Upon reviewing this work and performing the necessary calculations, we have identified several errors in the derived formulas. These inaccuracies affect the final results, particularly the growth rate of the Weibel instability in relation to the real and imaginary parts of the Weibel mode frequency. This comment aims to present these corrections in the order they appear in the original paper. To ensure clarity and reproducibility, detailed calculations are provided in three appendices.

## 2 Perturbed distribution function

The perturbed distribution function $\delta f^{\mp}$, derived from the Vlasov equation (Eq. (02) in the original manuscript), is given by

$$\delta f^{\mp} = i \frac{ebv_{\perp}}{mk} \left[ \frac{\partial F}{v_{\perp} \partial v_{\perp}} - \frac{\partial F}{v_{\parallel} \partial v_{\parallel}} + \frac{\left[ \omega \frac{\partial F}{v_{\parallel} \partial v_{\parallel}} \pm \omega_c \left( \frac{\partial F}{v_{\parallel} \partial v_{\parallel}} - \frac{\partial F}{v_{\perp} \partial v_{\perp}} \right) \right]}{\omega \pm \omega_c - k v_{\parallel}} \right].$$

This expression differs from Eq. (03) in the original paper. (See Appendix A for derivation details)

## 3 Current density

The current density $j_x$ is found to be

$$j_x = -i \frac{e^2 b \pi}{mk} \int v_{\perp}^3 dv_{\perp} dv_{\parallel} \left[ \frac{\partial F}{v_{\perp} \partial v_{\perp}} - \frac{\partial F}{v_{\parallel} \partial v_{\parallel}} + \frac{\left[ \omega \frac{\partial F}{v_{\parallel} \partial v_{\parallel}} \pm \omega_c \left( \frac{\partial F}{v_{\parallel} \partial v_{\parallel}} - \frac{\partial F}{v_{\perp} \partial v_{\perp}} \right) \right]}{\omega \pm \omega_c - k v_{\parallel}} \right].$$

This differs from Eq. (04) in the original paper in two ways:
1. A factor of ½ is missing.
2. The term $\pm \omega_c \left( \frac{\partial F}{v_{\parallel} \partial v_{\parallel}} - \frac{\partial F}{v_{\perp} \partial v_{\perp}} \right)$ has inverted signs

(See Appendix B for derivation details.)

## 4 Weibel instability analysis

Assuming $\frac{|\omega \pm \omega_c|}{k v_{\parallel}} \ll 1$, the imaginary part of the Weibel mode frequency $\text{Im}(\omega)$, is derived from the dispersion relation to be

$$Im(\omega) = \frac{k v_{T\parallel}}{\sqrt{\pi}} \frac{T_{\parallel}}{T_{\perp}} \left( -\frac{k^2 c^2}{\omega_p^2} - 1 + \frac{T_{\perp}}{T_{\parallel}} \right).$$

This differs from Eq. (9) in the original paper, where $k$ (a real positive wavenumber $k = \frac{2\pi}{\lambda}$) replaces $|k|$.

The real part of $\omega$ is given by

$$Re(\omega) = \pm \omega_c \frac{T_{\parallel}}{T_{\perp}} \left( -\left( \frac{T_{\perp}}{T_{\parallel}} - 1 \right) \left( 1 + \frac{2}{\pi} \frac{T_{\parallel}}{T_{\perp}} \right) + \frac{2}{\pi} \frac{T_{\parallel}}{T_{\perp}} \frac{k^2 c^2}{\omega_p^2} \right).$$

This expression differs from Eq. (11) in the original manuscript. (See Appendix C for derivation details).

From these corrected equations, we can derive the maximum wavenumber $k_{max}$ and the corresponding real frequency $Re(\omega)_{max}$:

$$k_{max} = \frac{\omega_p}{\sqrt{3}c} \sqrt{\left( \frac{T_{\perp}}{T_{\parallel}} - 1 \right)}$$

$$Re(\omega)_{max} = \mp \omega_c \left( 1 + \frac{4}{3\pi} \frac{T_{\parallel}}{T_{\perp}} \right) \left( 1 - \frac{T_{\parallel}}{T_{\perp}} \right).$$

For small temperature anisotropy, $\left( 1 - \frac{T_{\parallel}}{T_{\perp}} \right) \ll 1$, $Re(\omega)_{max} \ll \omega_c$.



While this expression for $Re(\omega)_{max}$ differs from Eq. (15) in the original paper, the conclusion remains unchanged: the Weibel mode remains a low-frequency instability even in the presence of nonzero external magnetic field.

## 5 Conclusion

In conclusion, an investigation of Weibel instability in magnetized plasma, is presented by Pokhotelov and Balikhin (Ann. Geophysicae, 30, 1051–1054, (2012)). In this reference, the parameters of Weibel instability are explicitly calculated under some justified approximations: i) The quasi-static modes approximation, ii) The linear approximation iii) The non-collisional approximation.

Upon reviewing this work and performing the necessary calculations, several errors in the derived formulas, especially in that of the Weibel mode frequency, $Re(\omega)$, are identified and corrected. With this correction the growth rate remains coincide with the classical expression. It contains a term in $k$, proportional to the $\frac{\Delta T}{T_\parallel}$ corresponds to the Weibel instability source and another term in $k^3$ due to the non-collisional absorption by Landau effect.

$Re(\omega)$ is proportional to the cyclotron frequency, $\omega_c$, and to $\frac{\Delta T}{T_\parallel}$: $Re(\omega) \sim \mp \omega_c \left(\frac{T_\perp}{T_\parallel} - 1\right) = \mp \omega_c \frac{T_\perp - T_\parallel}{T_\parallel} == \mp \omega_c \frac{\Delta T}{T_\parallel}$.

For the low magnetized plasma, the Weibel mode is purely growing electromagnetic mode , $Re(\omega) \approx 0$ , however for high magnetized plasma like that of magneto-inertial fusion the Weibel mode is convective.

### Acknowledgments

We are grateful to the laboratory of physics f radiations and their interactions with matter (PRIMALAB), University of Batna 1 for its support during this work..

### Authors contributions
 RN, AS and RM conceptualized the idea**;** AS supervised; RN, AS and NFF wrote the manuscript draft; RM and NFF reviewed and edited the manuscript.

## Declaration
The authors declare no competing interests.

### 6 References

Weibel, E. S.: Spontaneously growing transverse waves in a plasma due to an anisotropic velocity distribution, Phys. Rev. Lett., 2, 83–84, 1959.

Pokhotelov, O. A. and M. A. Balikhin, M. A.: Weibel instability in a plasma with nonzero external magnetic field, Ann. Geophys., 30, 1051–1054, 2012.

### Appendix A

The perturbed distribution function is calculated from the perturbed Vlasov equation (Eq. 1):

$$-i(\omega - kv_\parallel)\delta f_{\omega,k} + \omega_c \frac{\partial(\delta f_{\omega,k})}{\partial \alpha} = \frac{ebv_\perp}{mk}\left[(\omega - kv_\parallel)\frac{\partial F}{v_\perp \partial v_\perp} + k\frac{\partial F}{\partial v_\parallel}\right]\exp(-i\omega t + ikz)\exp(\mp i\alpha) \tag{A1}$$

This equation has a solution of the form:

$$\delta f_{\omega,k} = \delta f^{\mp}\exp(-i\omega t + ikz)\exp(\mp i\alpha). \tag{A2}$$





This implied that:

$$\frac{\partial(\delta f_{\omega,k})}{\partial \alpha} = \mp i \delta f_{\omega,k} \ . \tag{A3}$$

by substitution of Eq. (A2) in Eq. (A3), we obtain:

$$-i(\omega \pm \omega_c - k v_\parallel)\delta f^{\mp}\exp(-i\omega t + ikz)\exp(\mp i\alpha) = \frac{ebv_\perp}{mk}\left[(\omega - kv_\parallel)\frac{\partial F}{v_\perp \partial v_\perp} + k\frac{\partial F}{\partial v_\parallel}\right]\exp(-i\omega t + ikz)\exp(\mp i\alpha). \tag{A4}$$

By dividing (A4) by $\frac{\exp(-i\omega t + ikz)\exp(\mp i\alpha)}{-i(\omega \pm \omega_c - k v_\parallel)}$ , we obtain the expression of $\delta f^{\mp}$ as follow :

$$\delta f^{\mp} = \frac{ebv_\perp}{mk}\frac{\left[(\omega - kv_\parallel)\frac{\partial F}{v_\perp \partial v_\perp} + k\frac{\partial F}{\partial v_\parallel}\right]}{-i\omega \mp i\omega_c + ikv_\parallel} \ . \tag{A5}$$

This Eq. (A5) can be rewritten as:

$$\delta f^{\mp} = i\frac{ebv_\perp}{mk}\frac{\left[(\omega - kv_\parallel \pm \omega_c)\frac{\partial F}{v_\perp \partial v_\perp} \mp \omega_c\frac{\partial F}{v_\perp \partial v_\perp} - (\omega \pm \omega_c - kv_\parallel)\frac{\partial F}{v_\parallel \partial v_\parallel} + (\omega \pm \omega_c)\frac{\partial F}{v_\parallel \partial v_\parallel}\right]}{\omega \pm \omega_c - kv_\parallel} \tag{A6}$$

or again as:

$$\delta f^{\mp} = i\frac{ebv_\perp}{mk}\left[\frac{\partial F}{v_\perp \partial v_\perp} - \frac{\partial F}{v_\parallel \partial v_\parallel} + \frac{\left[\omega\frac{\partial F}{v_\parallel \partial v_\parallel} \pm \omega_c\left(\frac{\partial F}{v_\parallel \partial v_\parallel} - \frac{\partial F}{v_\perp \partial v_\perp}\right)\right]}{\omega \pm \omega_c - kv_\parallel}\right] \ . \tag{A7}$$

This expression of $\delta f^{\mp}$ is different from the expression of $\delta f^{\mp}$ given in the paper (Eq. 3).

**Appendix B**

The current density is given by:

$$j_x = -e\int v_x \delta f_{\omega,k}d^3\vec{v} \ , \tag{B1}$$

where $\delta f_{\omega,k}$ is given by Eq. (A2), $v_x = v_\perp \cos(\alpha)$ and $d^3\vec{v} = v_\perp dv_\perp dv_\parallel d\alpha$. Then:

$$j_x = -e\int v_\perp^2 dv_\perp dv_\parallel d\alpha \delta f^{\mp}\cos(\alpha)(\cos(\alpha) \mp i\sin(\alpha)) \ . \tag{B2}$$

By integrating this equation on $\alpha$, where $\int_0^{2\pi} \cos^2(\alpha) = \pi$ and $\int_0^{2\pi} \cos(\alpha)\sin(\alpha) = 0$, we otain:

$$j_x = -\frac{e}{2}\int v_\perp^2 dv_\perp dv_\parallel \delta f^{\mp} \ . \tag{B3}$$

By replacing Eq. (A7) in Eq. (B3), the current density is found as:



$$j_x = -i\frac{e^2 b\pi}{mk}\int v_\perp^3 dv_\perp dv_\parallel \left[\frac{\partial F}{v_\perp \partial v_\perp} - \frac{\partial F}{v_\parallel \partial v_\parallel} + \frac{\left[\omega\frac{\partial F}{v_\parallel \partial v_\parallel} \pm \omega_c\left(\frac{\partial F}{v_\parallel \partial v_\parallel} - \frac{\partial F}{v_\perp \partial v_\perp}\right)\right]}{\omega \pm \omega_c - kv_\parallel}\right].$$ (B4)

(Note here that we have ignored the multiplication by phase factor, $\exp(-i\omega t + ikz)$, which is the same in the all perturbed quantities.

This equation (B4) differs from Eq. (4) of the paper.

**Appendix C**

The dispersion relation is obtained from the Ampere's law, by ignoring the displacement current:
$$i\vec{k} \times \vec{\delta B} = \mu_0 \vec{j}.$$ (C1)

The y component of this equation is:

$$ikb = \mu_0 j_x.$$ C2)

By substitution by Eq. (B4) in this equation, we obtain:

$$k = \mu_0 \frac{e^2 \pi}{mk}\int v_\perp^3 dv_\perp dv_\parallel \left[\frac{\partial F}{v_\perp \partial v_\perp} - \frac{\partial F}{v_\parallel \partial v_\parallel} + \frac{\left[\omega\frac{\partial F}{v_\parallel \partial v_\parallel} \pm \omega_c\left(\frac{\partial F}{v_\parallel \partial v_\parallel} - \frac{\partial F}{v_\perp \partial v_\perp}\right)\right]}{\omega \pm \omega_c - kv_\parallel}\right].$$ (C3)

By regrouping the terms in (C3) and use the expressions of $\mu_0 = \frac{1}{c^2\varepsilon_0}$ and $\omega_p^2 = \frac{ne^2}{m\varepsilon_0}$, Eq. (C3) can be represented as:

$$\frac{k^2 c^2}{\omega_p^2} - \frac{\pi}{n}\int v_\perp^3 dv_\perp dv_\parallel \left[\frac{\partial F}{v_\perp \partial v_\perp} - \frac{\partial F}{v_\parallel \partial v_\parallel} + \frac{\left[\omega\frac{\partial F}{v_\parallel \partial v_\parallel} \pm \omega_c\left(\frac{\partial F}{v_\parallel \partial v_\parallel} - \frac{\partial F}{v_\perp \partial v_\perp}\right)\right]}{\omega \pm \omega_c - kv_\parallel}\right] = 0.$$ C4)

It differs from Eq. (5) of the paper by inversing the order of $\frac{\partial F}{v_\parallel \partial v_\parallel}$ and $\frac{\partial F}{v_\perp \partial v_\perp}$ in the term $\pm\omega_c\left(\frac{\partial F}{v_\parallel \partial v_\parallel} - \frac{\partial F}{v_\perp \partial v_\perp}\right)$.

By separation of terms in integral, Eq. (C4) can be rewrite as:

$$-\frac{k^2 c^2}{\omega_p^2} + \frac{\pi}{n}\int v_\perp^3 dv_\perp dv_\parallel \frac{\partial F}{v_\perp \partial v_\perp} - \frac{\pi}{n}\int v_\perp^3 dv_\perp dv_\parallel \frac{\partial F}{v_\parallel \partial v_\parallel} + \frac{\pi}{n}\int v_\perp^3 dv_\perp dv_\parallel \left[\frac{\left[\omega\frac{\partial F}{v_\parallel \partial v_\parallel} \pm \omega_c\left(\frac{\partial F}{v_\parallel \partial v_\parallel} - \frac{\partial F}{v_\perp \partial v_\perp}\right)\right]}{\omega \pm \omega_c - kv_\parallel}\right] = 0.$$ (C5)

The integrals can be evaluated, by considering the bi-maxwellain distribution (Eq. 8 of the paper), where: $\frac{\partial F}{v_\parallel \partial v_\parallel} = -\frac{2}{v_{T\parallel}^2}F$ and $\frac{\partial F}{v_\perp \partial v_\perp} = -\frac{2}{v_{T\perp}^2}F$.

- First integral:





$$I_1 = \int v_\perp^3 \, dv_\perp dv_\parallel \frac{\partial F}{v_\perp \partial v_\perp} = -\frac{2}{v_{T\perp}^2} \int_{-\infty}^{+\infty} dv_\parallel \int_0^\infty v_\perp^3 dv_\perp F = -\frac{n}{\pi}. \tag{C6}$$

• Second integral:

$$I_2 = \int v_\perp^3 dv_\perp dv_\parallel \frac{\partial F}{v_\parallel \partial v_\parallel} = -\frac{2}{v_{T\parallel}^2} \int v_\perp^3 dv_\perp dv_\parallel F = +\frac{n}{\pi}\frac{v_{T\perp}^2}{v_{T\parallel}^2} = \frac{n}{\pi}\frac{T_\perp}{T_\parallel}. \tag{C7}$$

• Third integral:

$$I3 = \int v_\perp^3 dv_\perp dv_\parallel \left[ \frac{\left[\omega \frac{\partial F}{v_\parallel \partial v_\parallel} \pm \omega_c \left(\frac{\partial F}{v_\parallel \partial v_\parallel} - \frac{\partial F}{v_\perp \partial v_\perp}\right)\right]}{\omega \pm \omega_c - k v_\parallel} \right]. \tag{C8}$$

By using the expressions of $F$ (Eq. 8), $\frac{\partial F}{v_\parallel \partial v_\parallel}$, and $\frac{\partial F}{v_\perp \partial v_\perp}$, the variables in the integral $I_3$ can be separated as follows:

$$I_3 = \frac{2N\pi}{nv_{T\perp}^2}\left[-\omega \frac{v_{T\perp}^2}{v_{T\parallel}^2} \pm \omega_c\left(1 - \frac{v_{T\perp}^2}{v_{T\parallel}^2}\right)\right]\int_{v_\perp=0}^{v_\perp=\infty} v_\perp^3 dv_\perp \exp(-\frac{v_\perp^2}{v_{T\perp}^2})\int_{v_\parallel=-\infty}^{v_\parallel=+\infty} dv_\parallel \left[\frac{\exp(-\frac{v_\parallel^2}{v_{T\parallel}^2})}{\omega \pm \omega_c - k v_\parallel}\right]. \tag{C9}$$

The integration on $v_\perp$ can be evaluated directely, so :

$$I_{3\perp} = \int_{v_\perp=0}^{v_\perp=\infty} v_\perp^3 dv_\perp \exp(-\frac{v_\perp^2}{v_{T\perp}^2}) = \frac{v_{T\perp}^4}{2}. \tag{C10}$$

The *plasma* dispersion *function* Z(s) is a fundamental complex special *integral function* widely used in the field of *plasma* physics

The integration on $v_\parallel$ is evaluated as follow :

$$I_{3\parallel} = \int_{v_\parallel=-\infty}^{v_\parallel=+\infty} dv_\parallel \left[\frac{\exp(-\frac{v_\parallel^2}{v_{T\parallel}^2})}{\omega \pm \omega_c - k v_\parallel}\right]. \tag{C11}$$

By using the variable changement : $y = \frac{v_\parallel}{v_{T\parallel}}$, $I_{32}$ can be presented as :

$$I_{3\parallel} = -\frac{1}{k}\int dy \left[\frac{\exp(-y^2)}{y - \frac{\omega \pm \omega_c}{k v_{T\parallel}}}\right]. \tag{C12}$$

This corresponds to the plasma dispersion function, $Z(\zeta)$:

$$I_{3\parallel} = -\frac{1}{k}\sqrt{\pi} Z(\zeta), \tag{C13}$$

with $\zeta = \frac{\omega \pm \omega_c}{k v_{T\parallel}}$



Then, the third integral, $I_3$, is presented as:

$$I_3 = -\frac{1}{kv_{T\parallel}}\left[-\omega\frac{v_{T\perp}^2}{v_{T\parallel}^2} \pm \omega_c\left(1-\frac{v_{T\perp}^2}{v_{T\parallel}^2}\right)\right]Z(\zeta).$$
(C14)

By substitution of equations (C6), (C7) and (C14) in Eq. (C5), the dispersion relation is obtained as::

$$-\frac{k^2c^2}{\omega_p^2} - 1 + \frac{T_\perp}{T_\parallel} + \frac{1}{kv_{T\parallel}}\frac{T_\perp}{T_\parallel}\left[\omega \mp \omega_c\frac{T_\parallel}{T_\perp}\left(\frac{T_\parallel}{T_\perp}-1\right)\right]Z(\zeta) = 0.$$
(C15)

By mean that $Z(\zeta) \approx i\sqrt{\pi} - 2\zeta$, Eq. (C14) rewrite as;

$$-\frac{k^2c^2}{\omega_p^2} - 1 + \frac{T_\perp}{T_\parallel} + \frac{1}{kv_{T\parallel}}\frac{T_\perp}{T_\parallel}\left[\frac{\omega\pm\omega_c}{v_{T\parallel}k} \mp \frac{\omega_c}{v_{T\parallel}k}\frac{T_\parallel}{T_\perp}\right]\left(i\sqrt{\pi} - 2\frac{\omega\pm\omega_c}{kv_{T\parallel}}\right) = 0,$$
C16)

where $\omega = Re(\omega) + iIm(\omega)$.

To compute $Im(\omega)$ and $Re(\omega)$, we separate the real and the imaginary part in Eq. (C16). The system of two coupled
equations is then obtained:

$$-\sqrt{\pi}\frac{T_\perp}{T_\parallel}\frac{Im(\omega)}{v_{T\parallel}k} \pm \frac{2\omega_c}{kv_{T\parallel}}\frac{Re(\omega)}{kv_{T\parallel}} = \frac{k^2c^2}{\omega_p^2} + 1 - \frac{T_\perp}{T_\parallel} - \frac{2\omega_c^2}{(kv_{T\parallel})^2}.$$
(C17)

$$\pm 2\frac{\omega_c}{kv_{T\parallel}}\frac{T_\parallel}{T_\perp}\frac{Im(\omega)}{kv_{T\parallel}} + \sqrt{\pi}\frac{Re(\omega)}{v_{T\parallel}k} = \pm\sqrt{\pi}\frac{\omega_c}{v_{T\parallel}k}\left(\frac{T_\parallel}{T_\perp}-1\right).$$
(C18)

This system can be presented as follow:

$$\begin{pmatrix} -\sqrt{\pi}\frac{T_\perp}{T_\parallel} & \pm\frac{2\omega_c}{kv_{T\parallel}} \\ \pm 2\frac{\omega_c}{kv_{T\parallel}}\frac{T_\parallel}{T_\perp} & \sqrt{\pi} \end{pmatrix}\begin{pmatrix} \frac{Im(\omega)}{v_{T\parallel}k} \\ \frac{Re(\omega)}{v_{T\parallel}k} \end{pmatrix} = \begin{pmatrix} \frac{k^2c^2}{\omega_p^2} + 1 - \frac{T_\perp}{T_\parallel} - \frac{2\omega_c^2}{(kv_{T\parallel})^2} \\ \pm\sqrt{\pi}\frac{\omega_c}{v_{T\parallel}k}\left(\frac{T_\parallel}{T_\perp}-1\right) \end{pmatrix}.$$
(C19)

This system has as a solution:

$$Im(\omega) = \frac{kv_{T\parallel}}{\sqrt{\pi}}\frac{T_\parallel}{T_\perp}\left(-\frac{k^2c^2}{\omega_p^2} - 1 + \frac{T_\perp}{T_\parallel}\right).$$
(C20)

$$Re(\omega) = \pm\omega_c\frac{T_\parallel}{T_\perp}\left(\left(1-\frac{T_\perp}{T_\parallel}\right)\left(1+\frac{2}{\pi}\frac{T_\parallel}{T_\perp}\right) + \frac{2}{\pi}\frac{T_\parallel}{T_\perp}\frac{k^2c^2}{\omega_p^2}\right).$$
(C21)

Note that the approximation $\left|\frac{\omega\pm\omega_c}{v_{T\parallel}k}\right| \ll 1$.

The $k_{max}$ is calculated from (C20) as:

$$k_{max} = \frac{\omega_p}{\sqrt{3}c}\sqrt{\left(\frac{T_\perp}{T_\parallel}-1\right)}.$$
(C22)

and the $Re(\omega)_{max}$ is calculated by substitution (C22) in (C21), so:





$$Re(\omega)_{max} = \mp\omega_c \left(1 + \frac{4}{3\pi}\frac{T_\parallel}{T_\perp}\right)\left(1 - \frac{T_\parallel}{T_\perp}\right) \sim \left(1 - \frac{T_\parallel}{T_\perp}\right) \ll 1 \,. \tag{C23}$$

It differs from the expression of $Re(\omega)_{max}$ found in the paper (Eq. 15) but the Weibel mode remains low frequency even in the presence of nonzero external magnetic field as he concluded in the paper.
