# Peer review of "Comment on "Weibel instability in a plasma with nonzero external magnetic field" by O. A. Pokhotelov and M. A. Balikhin, Ann. Geophysicae, 30, 1051–1054, (2012)"

_EGUsphere, 2025_

## Referee Comment (RC1)

**Report on egusphere-2025-1703**

Without expressing my opinion regarding the correctness or incorrectness of the expressions in Ann. Geophysicae, 30, 1051–1054, (2012), I should note that the derivation in the Appendix of the Comment is extremely inaccurate. Eq. (A1) is written for the Fourier components but contains the factor $\exp(-i\omega t + ikz)$, which should not be there. Eq. (A3) contains $\delta$ in the right-hand side. What is this $\delta$? Upon substitution of (A7) into (B3), the factor $1/2$ is lost. Substitution of (C14) into (C5) does not seem to be correct:

$$
-\omega \frac{v_{T\perp}^2}{v_{T\parallel}^2} \pm \omega_c \left(1 - \frac{v_{T\perp}^2}{v_{T\parallel}^2}\right)
$$
$$
= \frac{T_\perp}{T_\parallel}\left(-\omega \pm \omega_c \left(\frac{T_\parallel}{T_\perp} - 1\right)\right), \quad \text{Pokhotelov and Balikhin}
$$
$$
\neq \frac{T_\perp}{T_\parallel}\left(-\omega \pm \omega_c \frac{T_\parallel}{T_\perp}\left(\frac{T_\parallel}{T_\perp} - 1\right)\right), \quad \text{this comment}
$$

There is no reason to publish the comment.

---

## Author Comment (AC1)

**Response to Referee #1**

We would like to begin by expressing our appreciation to the Editor for their attention to our manuscript. We also extend our sincere thanks to Referee 1 for their review and comments. However, we would like to emphasize that the referee's remarks focused solely on typographical errors in the appendices, without addressing the validity of the comments concerning the original publication, as outlined in the main text of our comment.

Our response, on the comments of the Referee **#1,** is as follow:

- **Comment 1**

  Eq. (A1) is written for the Fourier components but contains the factor $exp(-i\omega t + ikz)$ which should not be there.

**Response 1**

We employed a direct calculation of the perturbed distribution function from the Vlasov equation and not the Fourier transform approach. The distribution function is assumed to consist of a bi-Maxwellian equilibrium component and a perturbation induced by the Weibel mode, characterized by frequency $\omega$ and wave number $k$: $\delta f(t, z, \vec{v})$. This perturbation is assumed to follow the spatio-temporal variation of the Weibel mode fields, in accordance with linear theory:

$\delta f(t, z, \vec{v}) = \delta f_{\omega,k}(\vec{v}) exp(ikz - i\omega t)$.

Substituting this form into the Vlasov equation reveals that $\delta f_{\omega,k}(\vec{v})$ exhibits angular dependence of the form $exp(\mp i\alpha)$.

Consequently, $\delta f_{\omega,k}(\vec{v})$ can be expressed as: $\delta f_{\omega,k}(\vec{v}) = \delta f^{\mp}(v_\parallel, v_\perp)exp(\mp i\alpha)$.

- **Comment 2**

  Eq. (A3) contains $\delta$ in the right-hand side. What is this $\delta$?

**Response 2**

The symbol $\delta$ is not interpreted in isolation, it is read together with $f_{\omega,k}$ as $\delta f_{\omega,k}$.

- **Comment 3**

Upon substitution of (A7) into (B3), the factor 1/2 is lost.

**Response 3**

This was merely a typographical error, where $\pi$ was mistakenly replaced with ½ in Eq. (B3). This error has no impact on the other equations or on the final results.

- **Comment 4**

  Substitution of (C14) into (C5) does not seem to be correct:

**Response 4**

Equation C14 was correctly substituted into Equation C5. The issue in Equation C15 was purely typographical, where the expression was mistakenly written as $\frac{T_\parallel}{T_\perp}\left(\frac{T_\parallel}{T_\perp} - 1\right)$ instead of the correct form $\left(\frac{T_\parallel}{T_\perp} - 1\right)$. This typographical error has no impact on the other equations or the final results, as the correct expression was used in all calculations. This can be easily verified.

**Dear Referee1,**

I respectfully disagree with the emphasis placed on typographical errors, in appendices, that neither affect the results nor undermine the main purpose of this publication. Since our comments manuscript requieres only minor corrections in the appendices, I kindly ask you to reconsider your opinion. Please find the corrected version attached.

[revised manuscript text omitted]
\frac{v_\perp^2}{v_\parallel^2} \pm \omega_c\left(1 - \frac{v_\perp^2}{v_\parallel^2}\right)\right]Z(\zeta). \tag{C14}$$

195

By substitution of equations (C6), (C7) and (C14) in Eq. (C5), the dispersion relation is obtained as::

$$-\frac{k^2c^2}{\omega_p^2} - 1 + \frac{T_\perp}{T_\parallel} + \frac{1}{kv_{T\parallel}}\frac{T_\perp}{T_\parallel}\left[\omega \mp \omega_c\left(\frac{T_\parallel}{T_\perp} - 1\right)\right]Z(\zeta) = 0. \tag{C15}$$

By mean that $Z(\zeta) \approx i\sqrt{\pi} - 2\zeta$, Eq. (C15) rewrite as;

$$-\frac{k^2c^2}{\omega_p^2} - 1 + \frac{T_\perp}{T_\parallel} + \frac{T_\perp}{T_\parallel}\left[\frac{\omega\pm\omega_c}{v_{T\parallel}k} \mp \frac{\omega_c}{v_{T\parallel}k}\frac{T_\parallel}{T_\perp}\right]\left(i\sqrt{\pi} - 2\frac{\omega\pm\omega_c}{kv_{T\parallel}}\right) = 0, \tag{C16}$$

200 where $\omega = Re(\omega) + iIm(\omega)$.

To compute $Im(\omega)$ and $Re(\omega)$, we separate the real and the imaginary part in Eq. (C16). The system of two coupled equations is then obtained:

$$-\sqrt{\pi}\frac{T_\perp}{T_\parallel}\frac{Im(\omega)}{v_{T\parallel}k} \pm \frac{2\omega_c}{kv_{T\parallel}}\frac{Re(\omega)}{kv_{T\parallel}} = \frac{k^2c^2}{\omega_p^2} + 1 - \frac{T_\perp}{T_\parallel} - \frac{2\omega_c^2}{\left(kv_{T\parallel}\right)^2}. \tag{C17}$$

$$\pm 2\frac{\omega_c}{kv_{T\parallel}}\frac{T_\parallel}{T_\perp}\frac{Im(\omega)}{kv_{T\parallel}} + \sqrt{\pi}\frac{Re(\omega)}{v_{T\parallel}k} = \pm\sqrt{\pi}\frac{\omega_c}{v_{T\parallel}k}\left(\frac{T_\parallel}{T_\perp} - 1\right). \tag{C18}$$

205 This system can be presented as follow:

$$\begin{pmatrix} -\sqrt{\pi}\frac{T_\perp}{T_\parallel} & \pm\frac{2\omega_c}{kv_{T\parallel}} \\ \pm 2\frac{\omega_c}{kv_{T\parallel}}\frac{T_\parallel}{T_\perp} & \sqrt{\pi} \end{pmatrix}\begin{pmatrix} \frac{Im(\omega)}{v_{T\parallel}k} \\ \frac{Re(\omega)}{v_{T\parallel}k} \end{pmatrix} = \begin{pmatrix} \frac{k^2c^2}{\omega_p^2} + 1 - \frac{T_\perp}{T_\parallel} - \frac{2\omega_c^2}{\left(kv_{T\parallel}\right)^2} \\ \pm\sqrt{\pi}\frac{\omega_c}{v_{T\parallel}k}\left(\frac{T_\parallel}{T_\perp} - 1\right) \end{pmatrix}. \tag{C19}$$

This system has as a solution:

$$Im(\omega) = \frac{kv_{T\parallel}}{\sqrt{\pi}}\frac{T_\parallel}{T_\perp}\left(-\frac{k^2c^2}{\omega_p^2} - 1 + \frac{T_\perp}{T_\parallel}\right). \tag{C20}$$

$$Re(\omega) = \pm\omega_c\frac{T_\parallel}{T_\perp}\left(\left(1 - \frac{T_\perp}{T_\parallel}\right)\left(1 + \frac{2}{\pi}\frac{T_\parallel}{T_\perp}\right) + \frac{2}{\pi}\frac{T_\parallel}{T_\perp}\frac{k^2c^2}{\omega_p^2}\right). \tag{C21}$$

210 Note that the approximation $\left|\frac{\omega\pm\omega_c}{v_{T\parallel}k}\right| \ll 1$ is used.

The $k_{max}$ is calculated from (C20) as:

$$k_{max} = \frac{\omega_p}{\sqrt{3}c}\sqrt{\left(\frac{T_\perp}{T_\parallel} - 1\right)} . \tag{C22}$$

and the $Re(\omega)_{max}$ is calculated by substitution (C22) in (C21), so:

$$Re(\omega)_{max} = \mp\omega_c \left(1 + \frac{4}{3\pi}\frac{T_\parallel}{T_\perp}\right)\left(1 - \frac{T_\parallel}{T_\perp}\right) \sim \left(1 - \frac{T_\parallel}{T_\perp}\right) \ll 1 \ . \tag{C23}$$

215   It differs from the expression of $Re(\omega)_{max}$ found in the paper (Eq. 15) but the Weibel mode remains low frequency even in the presence of nonzero external magnetic field as he concluded in the paper.

---

## Author Comment (AC2)

First of all, I would like to thank Referee #1 for their understanding of my response to the previous report.

According to the second referee's report, the mistakes identified in comment pertain solely to the intermediate formulas and do not affect the final result. In his respected opinion, the result—namely the dispersion relation—remains the same as the one derived in the comment.

The response on the second report of Referee #1 is as follow:
Mistakes in the origin paper by Pokhotelov and Balikhin are well indicated in the main text of our comment paper. These mistakes are not only in the intermediate steps of calculation (Eqs. 3 and 4) but also in the final results relatives to the parameters of Weibel instability which is the subject of the original paper:

- The derived expression for $Re(\omega)$, which defines the oscillation frequency of the Weibel mode, significantly differs from the formulation reported in Eq. (13) of the original paper by Pokhotelov and Balikhin

$$Re(\omega) = \pm\omega_c \frac{T_\parallel}{T_\perp}\left(-\left(\frac{T_\perp}{T_\parallel}-1\right)\left(1+\frac{2}{\pi}\frac{T_\parallel}{T_\perp}\right)+\frac{2}{\pi}\frac{T_\parallel}{T_\perp}\frac{k^2c^2}{\omega_p^2}\right)$$ in our comment

$$Re(\omega) = \pm\omega_c \frac{T_\parallel}{T_\perp}\left(\left(\frac{T_\perp}{T_\parallel}-1\right)\left(1+\frac{2}{\pi}\frac{T_\parallel}{T_\perp}\right)+\frac{2}{\pi}\frac{T_\parallel}{T_\perp}\frac{k^2c^2}{\omega_p^2}\right)$$ in Pokhotelov and Balikhin

- The expression for $Im(\omega)$ in the original paper (Eq. 12), representing the growth rate of the Weibel mode, contains mistakes: Pokhotelov and Balikhin use $|k|$ instead of $k$, despite the fact that the wave number is defined in instability theory as a positive real quantity.
- The expression for $Re(\omega)_{max}$ in Eq. (15) of the original paper, representing the oscillation frequency of most unstable Weibel mode, contains errors:

$$Re(\omega)_{max} = \mp\omega_c\left(1+\frac{4}{3\pi}\frac{T_\parallel}{T_\perp}\right)\left(1-\frac{T_\parallel}{T_\perp}\right)$$  in comment

$$Re(\omega)_{max} = \pm 058\omega_c\left(1-\frac{T_\parallel}{T_\perp}\right)$$  in Pokhotelov and Balikhin

-The expression obtained in comment is clearly different than that in the original paper.
-The sign is inverted: $\mp$ in our comment but $\pm$ in the original paper (- for the right polarization and + for the left one).

- The dispersion relation obtained in the original paper (Eq. 09) is not similar to that obtained in our comment (C15, C16) by using $|k|$ in place of $k$:
-this is a calculation mistakes
-the introduction of $|k|$ is not in agreement with the temporal instability where the $k$ is treated as a positive real and $\omega$ as a complex:

$$-\frac{k^2c^2}{\omega_p^2} - 1 + \frac{T_\perp}{T_\parallel} + \frac{1}{kv_{T\parallel}}\frac{T_\perp}{T_\parallel}\left[\omega \mp \omega_c\left(\frac{T_\parallel}{T_\perp}-1\right)\right]Z(\zeta) = 0.$$        Eq. (C15) of the comment

$$-\frac{k^2c^2}{\omega_p^2} - 1 + \frac{T_\perp}{T_\parallel} + \frac{T_\perp}{T_\parallel}\left[\frac{\omega\pm\omega_c}{v_{T\parallel}k} \mp \frac{\omega_c}{v_{T\parallel}k}\frac{T_\parallel}{T_\perp}\right]\left(i\sqrt{\pi}-2\frac{\omega\pm\omega_c}{kv_{T\parallel}}\right) = 0,$$        Eq. (C16) of the comment

$$\frac{T_\perp}{T_\parallel} - 1 - \frac{k^2c^2}{\omega_p^2} + \frac{T_\perp}{T_\parallel}\left[\frac{\omega\mp\omega_c\left(\frac{T_\parallel}{T_\perp}-1\right)}{|k|v_{T\parallel}}\right]Z(\frac{\omega\pm\omega_c}{|k|v_{T\parallel}}) = 0,$$        Eq. (09) of the original paper